# Using scalable computer vision to automate high-throughput semiconductor characterization

Alexander E. Siemenn [1,4] ✉, Eunice Aissi [1,4] ✉, Fang Sheng[1], Armi Tiihonen[1,2], Hamide Kavak [1,3], Basita Das[1] & Tonio Buonassisi[1]

High-throughput materials synthesis methods, crucial for discovering novel functional materials, face a bottleneck in property characterization. These high-throughput synthesis tools produce $10^4$ samples per hour using ink-based deposition while most characterization methods are either slow (conventional rates of $10^1$ samples per hour) or rigid (e.g., designed for standard thin films), resulting in a bottleneck. To address this, we propose automated characterization (autocharacterization) tools that leverage adaptive computer vision for an 85x faster throughput compared to non-automated workflows. Our tools include a generalizable composition mapping tool and two scalable autocharacterization algorithms that: (1) autonomously compute the band gaps of 200 compositions in 6 minutes, and (2) autonomously compute the environmental stability of 200 compositions in 20 minutes, achieving 98.5% and 96.9% accuracy, respectively, when benchmarked against domain expert manual evaluation. These tools, demonstrated on the formamidinium (FA) and methylammonium (MA) mixed-cation perovskite system $FA_{1-x}MA_xPbI_3$, $0 \le x \le 1$, significantly accelerate the characterization process, synchronizing it closer to the rate of high-throughput synthesis.

Automating and accelerating the characterization of materials is a challenge in the high-throughput materials discovery pipeline. Recent advancements have notably accelerated the synthesis of materials, playing a pivotal role in the discovery process. These synthesis methods have been developed across a wide range of material domains including perovskite semiconductors[1–6], nanomaterials[7], porous media[8], aerosols[9], and lithium-ion batteries[10]. Techniques such as inkjetting[11–13] and drop-casting[14,15] are pivotal in enabling high-throughput synthesis and screening of optimized functional materials[3,7,16]. However, these techniques produce droplet-like samples with variable morphologies, distinct in structural and geometric aspects from standard thin films[3,14,17], due to the inherent nature of the droplet deposition process[14,15,18]. Consequently, these droplet morphologies are not readily compatible with existing characterization tools, which are designed for uniform thin film analysis[1,2,6,19,20]. Therefore, to bridge the gap in throughput between accelerated synthesis and slow characterization, new adaptive and automated characterization tools must be designed to keep pace with synthesis rates while handling variable sample morphologies[21]. Bridging this gap is critical for the efficient discovery of novel materials.

High-throughput (HT) synthesis tools are capable of producing materials at rates approximately 800× faster than what existing characterization methods can handle (illustrated in Supplementary Fig. S-1)[16]. This bottleneck in characterization is due to the manual nature of these tools[6,22] and their inability to parallelize measurements across many samples of varying morphology[1,2,23–25]. Optoelectronic materials such as semiconductors highlight this challenge because of their complex and extensive material search

[1]Department of Mechanical Engineering, Massachusetts Institute of Technology, 77 Massachusetts Avenue, Cambridge 02139 MA, USA. [2]Department of Applied Physics, Aalto University, Otakaari 24, Espoo 02150, Finland. [3]Department of Physics, Cukurova University, Adana 01330, Turkey. [4]These authors contributed equally: Alexander E. Siemenn, Eunice Aissi. ✉e-mail: asiemenn@mit.edu; eunicea@mit.edu

space[1,6,19,26–28]. For semiconductors, their performance and commercial viability[29,30] hinge on accurately characterizing key properties like band gap[31–33] and environmental stability[19,20]. While recent studies have made progress in accelerating semiconductor characterization, a significant gap remains in developing automated and adaptive characterization tools that can keep pace with the rapid output of HT synthesis platforms. For example, Escobedo et al.[34] develop automated programs to compute the band gap of semiconductors, one material at a time, using pre-collected optical data. Surmiak et al.[24] and Reinhardt et al.[25] expand on this method by employing HT optical characterization of perovskites for up to 24 unique samples per batch but require the measurement positions to be statically hard-coded and occur in serial, in turn, lacking scalability and generalizability. Similarly, Langner et al.[2] and Wang et al.[1] develop significantly higher throughput liquid handling tools capable of synthesizing and characterizing up to 6048 organic photovoltaic materials per day by depositing solutions into microplates, while Du et al.[23] develop an automated robotic synthesis and characterization platform for full organic photovoltaic devices. These developed methods significantly advance semiconductor screening and discovery, however, these prior works each utilize rigid synthesis and characterization techniques that assume invariable sample geometry and placement, in turn, limiting the ultimate characterization throughput. Unlike the aforementioned literature, Wu et al.[21] develop HT and scalable tools, automated from end-to-end, to characterize the optical properties of organic molecules actively suspended in solution. Although Wu et al.[21] provide an impressive framework for synchronizing automated synthesis with characterization, the methodology does not apply to more general deposited materials. Thus, the task of rapidly and automatically characterizing semiconductors deposited with variable morphologies remains an open research gap.

Debottlenecking the materials screening pipeline is not only a matter of accelerating the characterization time per sample[18,35] but also a matter of scaling the characterization procedure to many samples in parallel[36,37]. Computer vision methods have the capacity to scale a measurement to arbitrarily many samples, each with differing form factor geometries, without significantly slowing characterization time[38,39]. Material science applications of computer vision are gaining traction in current literature, specifically in the use case of rapid morphological analysis of large microscopy datasets[39]. In turn, several analytical computer vision tools have been developed to access and characterize this morphological information, often focused on identifying microstructures. For example, Park et al.[40] create a semi-automated image segmentation and analysis algorithm to classify the morphology of nanoparticles from image data. Likewise, Chowdhury et al.[41] utilize computer vision and machine learning to detect dendritic microstructures from a database of solder alloy micrographs. There is also extensive work surrounding materials recognition in non-micrographic images using computer vision[18,42,43]. In order to support the growing need for the analysis of micrographs and other experimental images, advances like those made by Li et al.[38], Wang et al.[44], Neshatavar et al.[45], Tung et al.[36], and Jain et al.[37] on object segmentation, denoising, and scalability allow for further use cases of computer vision in scientific research. The throughput dichotomy between characterization and HT synthesis motivates the integration of computer vision into the semiconductor characterization pipeline to parallelize measurements and, in turn, match or exceed the rate of synthesis while achieving accuracies comparable to those attained by domain expert evaluation.

In this paper, we address the unresolved challenge of characterizing deposited materials quickly and automatically in parallel by developing a suite of computer vision-based automated characterization (autocharacterization) tools that adaptively quantify three key material properties within minutes: composition[46,47], optical band gap[31–33], and environmental degradation[1,19,20], as shown in Fig. 1. We propose the following in this contribution: (1) sample detection by a scalable computer vision tool that segments arbitrarily many, spatially non-uniform material samples, (2) a tool to map the elemental composition of HT-manufactured material arrays, (3) a scalable autocharacterization algorithm for the computation of direct band gaps from hyperspectral reflectance data, and (4) a scalable autocharacterization algorithm for quantifying the environmental stability of perovskite samples from optical degradation data. The performances of the developed autocharacterization methods are demonstrated on 200 unique HT-manufactured formamidinium (FA) and methylammonium (MA) mixed-cation lead halide perovskite semiconductor samples $FA_{1-x}MA_xPbI_3$, generating ultra-high compositional resolution trends of band gap and stability (available in Supplementary Data 1), and are benchmarked against X-ray diffraction (XRD)[48,49], X-ray photoelectron spectroscopy (XPS)[50,51], and domain expert evaluation.

## Results

### Computer vision parallelization

With the integration of computer vision into the materials characterization workflow, data across many samples can be captured and analyzed in parallel as a fast and scalable process[36–39]. Computer vision can be applied to both standard RGB and hyperspectral image data types. Figure 2 illustrates the segmentation of a hyperspectral datacube, taken from one batch of HT-manufactured perovskite samples[18]. In this study, we synthesize three batches of samples, amounting to a total of $N = 201$ unique semiconductors along the $FA_{1-x}MA_xPbI_3$ compositional series with $0 \leq x \leq 1$. Using Algorithm 1, we generate unique pairings for each discrete semiconductor sample, $(\widehat{X}, \widehat{Y})_n \in N$, and its corresponding reflectance spectra, $R(\lambda)$, via parallel image segmentation and mapping, shown in Fig. 2b.

The process of parallel segmentation uses a sequence of edge-detection filters[52] to first identify each island of material and then uniquely index each island based on its position within a graph connectivity network[53]. The pixel coordinates of each segmented sample are then spatially mapped to their corresponding reflectance spectra, in turn, generating the segmented datacube $\Phi = (\widehat{X}, \widehat{Y}, R(\lambda))$. Parallel segmentation and mapping of these reflectance spectra accelerate a once point-by-point measurement process[6,22], to a rapid and scalable process that can rate-match the throughput of HT synthesis (Supplementary Fig. S-1). Furthermore, this segmentation process is shown to scale to more than 80 unique samples in parallel (Supplementary Fig. S-4). Algorithm 1 illustrates the sample size-agnostic nature of the method, highlighting its further scalable potential. These segmented reflectance data serve as the starting point for automating the mapping of composition and the computation of optical band gap and degradation for all $N = 201$ unique semiconductors.

### Composition mapping

Semiconductor properties such as band gap and stability are largely governed by the chemical composition of the material[19,20,46]. Thus, accurately determining and mapping composition to each sample is essential for scalable HT setups that deposit variable sample geometries. In this work, we use HT deposition of solution-processed precursors to synthesize perovskite semiconductor samples. The variable rotational velocities of two pumps, $\omega_{FA}$ and $\omega_{MA}$, determine the output sample composition by combining the liquid-based formamidinium lead iodide ($FAPbI_3$) and methylammonium lead iodide ($MAPbI_3$) precursors. Figure 3a illustrates this printing process as a function of both space and time. The FA-rich materials are printed first and then proportions of MA are added gradually as the print head rasters in a

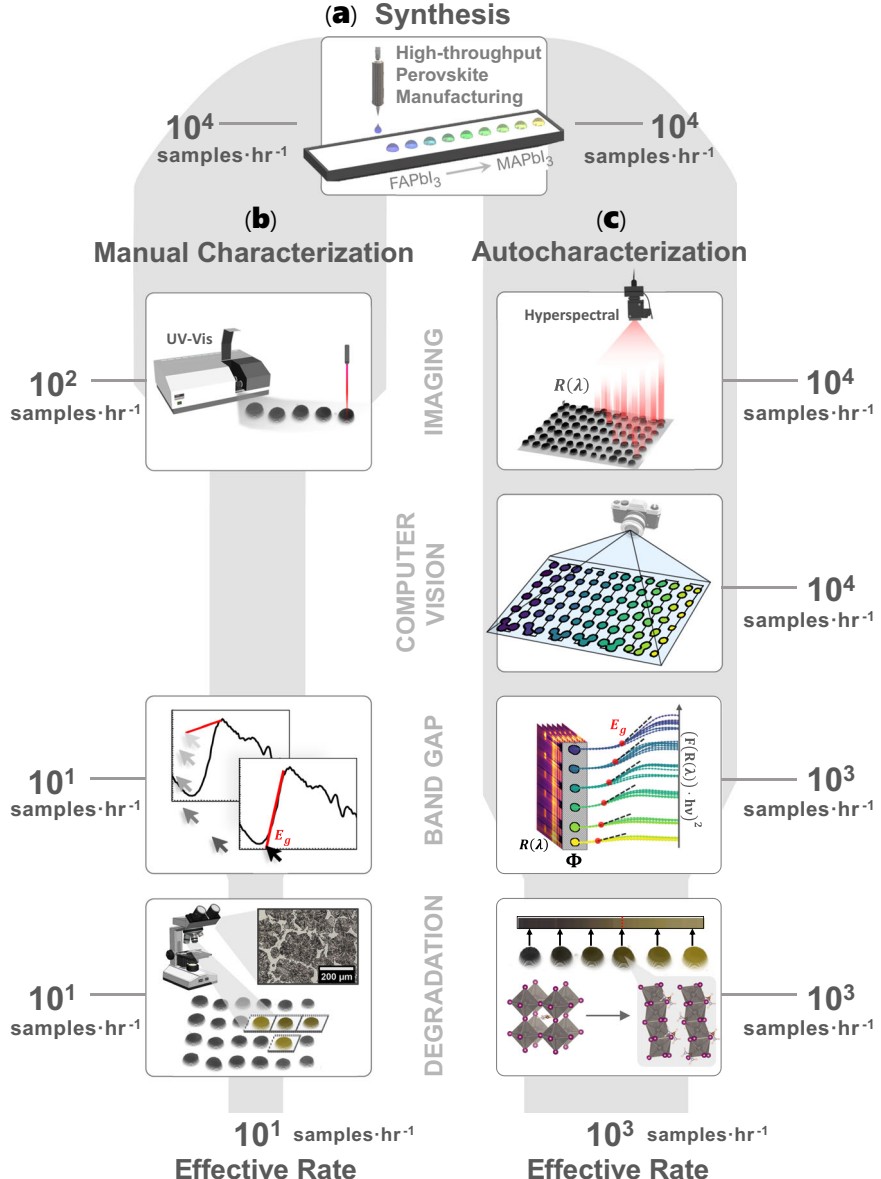

**Fig. 1 | Overview of the synthesis and characterization pipeline for perovskite semiconductors. a** High-throughput combinatorial synthesis of formamidinium (FA) and methylammonium (MA) mixed-cation perovskites $FA_{1-x}MA_xPbI_3$ attain throughputs of $10^4$ samples $h^{-1}$. **b** Manual characterization of the high-throughput-manufactured materials using UV–Vis spectroscopy and manual determination of band gap and degradation bottlenecks the pipeline down to a throughput of $10^1$ samples $h^{-1}$. **c** Autocharacterization, developed in this paper, of the high-throughput-manufactured material's band gaps and degradation attain throughputs of $10^3$ samples $h^{-1}$ using scalable and parallelizable computer vision measurement. Band gap is determined by automatically segmenting and fitting the material reflectance spectra while the degradation pathway is detected by the material yellowing, in the case depicted above due to a phase change from $\alpha$-FAPbI$_3$ to $\delta$-FAPbI$_3$. The widths of the gray backgrounds visualize the process throughputs.

serpentine pattern. Thus, to determine the composition of each material deposit, $\omega_{FA}$ and $\omega_{MA}$ are spatially and temporally mapped onto the computer vision-segmented samples, $(\hat{\mathbf{X}}, \hat{\mathbf{Y}})_n \in N$, and then integrated over time to determine each sample's computed $FA_{1-x}MA_xPbI_3$ composition:

$$x(t) \approx \int_{t_a}^{t_b} \frac{\omega_{MA}(t)}{(\omega_{MA}(t) + \omega_{FA}(t))} dt, \qquad (1)$$

where $x$ is the proportion of MA, $t_a$ and $t_b$ are the starting and ending timesteps for the deposition of a single sample, and $\omega_{FA}(t)$ and $\omega_{MA}(t)$ are the pump velocities at a given timestep for the FAPbI$_3$ and MAPbI$_3$ precursors, respectively.

Figure 3b, c illustrates the composition validation results using XRD and XPS. XRD is used to validate the crystalline structure[48,49] while XPS is used to validate the elemental composition of the manufactured perovskite deposits[50,51]. By assessing the gradated shifts in both XRD and XPS peaks, the composition mapping can be validated. For crystal structure validation, the crystal lattice size of MAPbI$_3$ is smaller than that of FAPbI$_3$[47], thus, we expect the XRD peaks of the MA-rich deposits to shift toward higher angles[54]. In Fig. 3b, the crystallographic plane at $2\theta \approx 31.5°$[55,56] is shown, for uniformly spaced samples across the batch print, to gradually increase in angle from the FA-rich to the MA-rich compositions of $FA_{1-x}MA_xPbI_3$, amounting to a total shift of 0.16° (quantitative shifts shown in Supplementary Fig. S-15a). For elemental validation, the A-site MA and FA cations are distinguished by the

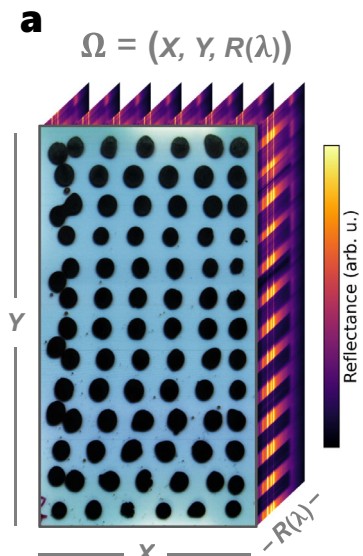

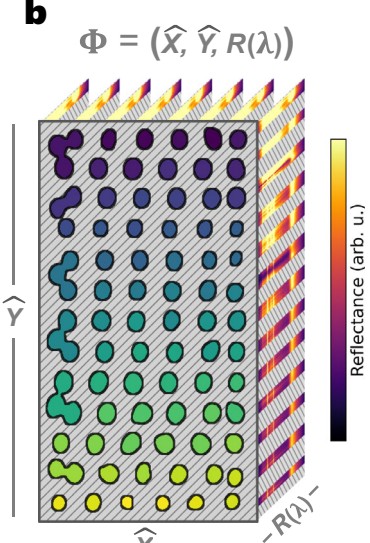

**Fig. 2 | Computer vision segmentation of hyperspectral data from high-throughput (HT) synthesized semiconductors. a** Raw hyperspectral datacube, **Ω**, captured using a hyperspectral imager (Resonon, Pika L) of HT-deposited for-mamidinium (FA) and methylammonium (MA) mixed-cation perovskites $FA_{1-x}MA_xPbI_3$. (**X**, **Y**) represents the pixel coordinates, and $R(\lambda)$ represents the reflectance spectra for each pixel. Each sample is deposited onto the glass substrate with a unique composition $0 \leq x \leq 1$ and flexible form factor geometry. **b** Computer-vision segmented datacube, **Φ**, that pairs each unique sample's pixels, $(\hat{X}, \hat{Y})_n \in N$, to its reflectance spectra, **R**$(\lambda)$. The gray hatched region indicates the discarded background pixels.

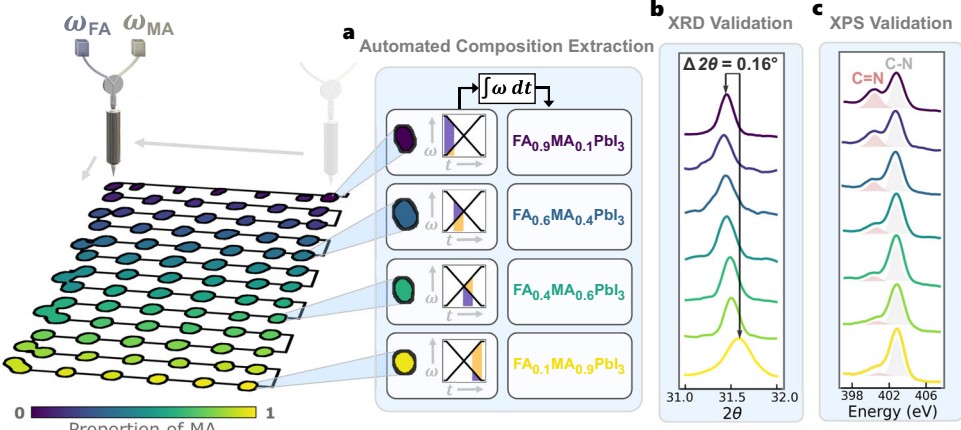

**Fig. 3 | High-throughput combinatorial synthesis of a batch of perovskite samples with corresponding validation measurements. a** The print head rasters in a serpentine pattern (black connecting lines) to print a gradient of for-mamidinium (FA) and methylammonium (MA) mixed-cation perovskite $FA_{1-x}MA_xPbI_3$ deposits onto a glass substrate with varying pump speeds, $\omega$. Integrating the pump speeds over time, $t$, determines the proportion of MA, $x$, in the composition. **b** X-ray diffraction (XRD) peak traces at the crystallographic plane with the Miller indices (012), measured at uniformly spaced compositions in the batch print. The peak shifts toward a higher total diffraction angle, $2\theta$, as the proportion of MA increases in the composition. **c** X-ray photoelectron spectroscopy (XPS) traces of the C=N bond peak (red area under the curve) and C−N bond peak (gray area under the curve) measured at uniformly spaced compositions in the batch print. The C=N peak intensity decreases as the proportion of MA increases. Source data are provided as a Source Data file.

presence of a carbon−nitrogen double bond (C=N), where FA contains a C=N bond while MA contains only a C−N single bond[54]. In the high-resolution XPS scans, shown in Fig. 3c, the C=N bond peak appears at approximately 400 eV[57]. This C=N bond peak is shown, for uniformly spaced samples across the batch print, to gradually decrease in intensity from the FA-rich to the MA-rich compositions of $FA_{1-x}MA_xPbI_3$ (quantitative shifts shown in Supplementary Fig. S-15b). These results validate both the structural and elemental composition gradients synthesized and mapped using computer vision. With the developed composition map, we can now automatically compute the band gap and detect degradation across the 201 uniquely synthesized compositions.

## Automated band gap extraction

Band gap is essential in defining the light-harvesting potential of a semiconductor material[31,47]. However, conventionally computing the band gap of a material is a laborious process, requiring a user to manually curve-fit the Tauc-transformed UV−Vis spectra[22,58]. In this paper, we automate and accelerate the computation of band gap across 201 unique semiconductor samples by leveraging hyperspectral imaging and computer vision segmentation such that the characterization process is parallelized across batches of HT-manufactured semiconductor samples.

Figure 4 illustrates the autocharacterization process of band gap by first (a) extracting the reflectance spectra from each computer

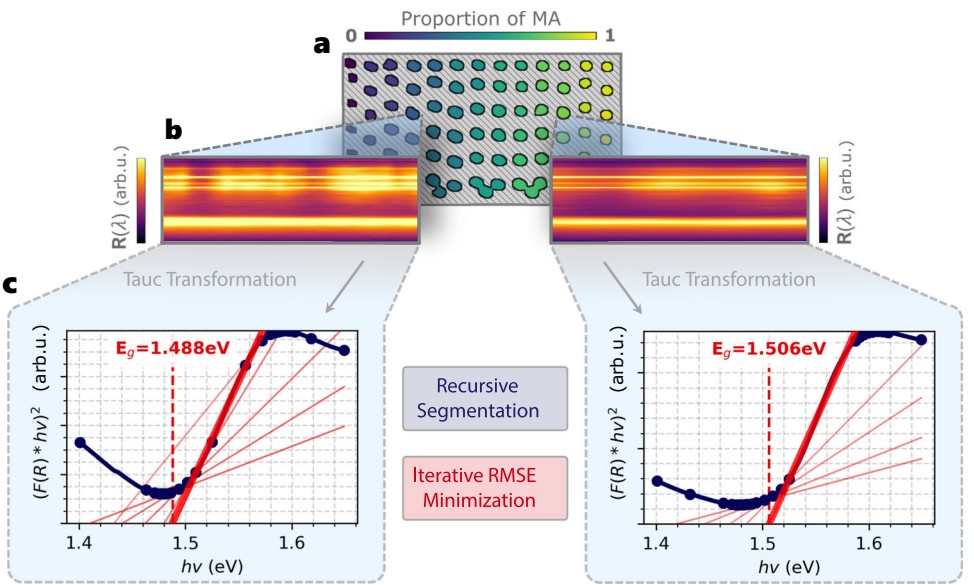

**Fig. 4 | Automated band gap computation shown for two unique computer vision-segmented perovskite deposits. a** Computer vision-segmented reflectance hypercube, $\mathbf{\Phi}$, for one batch of materials with increasing proportions of methylammonium (MA). **b** The reflectance intensities, $\mathbf{R}(\lambda)$, of perovskite deposits acquired from the hypercube. **c** The Tauc curves are computed from the median reflectance spectra for each deposit, recursively segmented into line segments, and then iteratively fit with linear regression lines. $F(\mathbf{R})$ is the Kubelka–Munk function, $h$ is the Planck constant, and $\nu$ is photon frequency. The best-fit regression line that minimizes the root-mean-square error (RMSE) between the detected Tauc peaks is illustrated by the thick red line, which determines the band gap, $E_g$, from the $x$-intercept.

vision-segmented sample within the hyperspectral datacube, then (b) transforming each reflectance spectra to its Tauc curve, and finally (c) recursively segmenting the Tauc curves into linear segments ($R^2 \geq 0.990$) to find the regression line of best fit between peaks, which determines band gap. The Tauc curves are obtained from a hyperspectral reflectance datacube using the following transformation[22]:

$$F(\mathbf{R}(\lambda) \cdot h\nu)^{1/\gamma} = B(h\nu - E_g), \quad (2)$$

where $F(\mathbf{R}(\lambda))$ is the Kubelka–Munk function[59] applied to reflectance spectra, $\mathbf{R}(\lambda)$, for each wavelength, $\lambda$, with $h$ as the Planck constant, $\nu$ as the photon frequency, B as a constant, and $\gamma = \frac{1}{2}$ for direct band gap and $\gamma = 2$ for indirect band gap.

To demonstrate the performance of the band gap autocharacterization developed in this paper, the algorithm's output band gaps are compared with those calculated by a domain expert using the manual fitting process described in ref. 22. The autocharacterization output was withheld from the domain expert. Figure 5 illustrates the performance of the autocharacterization-calculated band gaps relative to the expert-calculated band gaps for $N = 201$ FA$_{1-x}$MA$_x$PbI$_3$ compositions across three independent batches of samples. The autocharacterization output achieves a strong linear fit of $R^2 = 0.975$ with the expert-calculated results, however, a systematic underprediction of the autocharacterization algorithm is noted. Relative to the expert-computed band gap, the autocharacterization method achieves 98.5% accuracy within a 0.02 eV range on the FA$_{1-x}$MA$_x$PbI$_3$ system (Supplementary Fig. S-7). In addition to the autocharacterization achieving high similitude with the domain expert, it also provides significant speedups in band gap determination. The domain expert takes approximately 510 min to compute the band gap of 200 unique samples while the autocharacterization takes 6 min to compute the band gap of 200 samples, resulting in the developed autocharacterization tool achieving 85× faster throughput via the parallel-processing power of computer vision.

Using the fast and accurate band gap autocharacterization tool developed in this paper, we generate an ultra-high resolution band gap

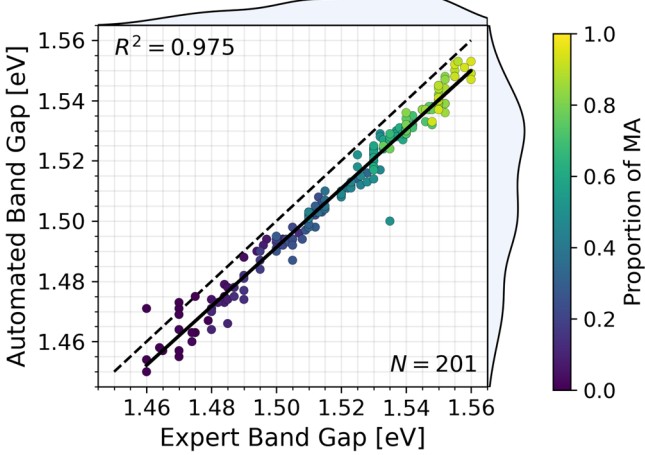

**Fig. 5 | Band gap autocharacterization algorithm performance compared against domain expert evaluation.** The band gap computations shown are for $N = 201$ unique perovskite samples across three independent trials. The solid black line is the line of best fit to the band gap data and the dashed black line is the $y = x$ line. Histogram distributions of both autocharacterization and domain expert band gaps are shown on the right and top of the plot area, respectively. The color of the scatter points corresponds to the proportion of MA, $x$, in the formamidinium (FA) and methylammonium (MA) mixed-cation composition FA$_{1-x}$MA$_x$PbI$_3$. Source data are provided as a Source Data file.

trend for the FA$_{1-x}$MA$_x$PbI$_3$, $0 \leq x \leq 1$ series, shown in Fig. 5, where 120 of the 201 compositions are unique, a resolution that has not yet been reported in literature to our knowledge. Prior literature reports band gap compositional resolutions from $0 \leq x \leq 1$ for 9 compositions[60], 7 compositions[61], and 5 compositions[19] using conventional characterization methods. Furthermore, with this high-resolution band gap trend, we achieve the same band gap values of 1.46 eV for FAPbI$_3$ and 1.55 eV for MAPbI$_3$, corresponding with the linear trends from literature[19,60,61]. Thus, with autocharacterization, we accurately achieve

over a 13× increase in the compositional resolution of FA$_{1-x}$MA$_x$PbI$_3$ band gap, to our knowledge.

## Automated degradation detection

Sufficient stability of perovskite semiconductors is required for the material to be utilized in solar cell applications[29,30,62]. As a lead halide perovskite degrades, it changes color from black to yellow, a result of a phase change and/or decomposition of the structure[19,63,64]. We leverage this RGB-detectable degradation mechanism[20] and parallelized computer vision segmentation to automate the detection of degradation within perovskites, as shown in Fig. 6c. Three independent degradation experiments are conducted across the $N = 201$ samples by placing each batch of samples within a degradation chamber, shown in Fig. 6a, for 2 h at an illumination of 0.5 suns, temperature of 34.5 ± 0.5 °C, and relative humidity of 40% ± 1% (Supplementary Fig. S-8). We compute the degradation intensity, $I_c$, of each HT-manufactured

perovskite composition by integrating the change in color, $R$, for each sample over time, $t$[19]:

$$I_c(\widehat{\mathbf{X}}, \widehat{\mathbf{Y}}) = \sum_{\mathbf{R} = \{r,g,b\}} \int_0^T |\mathbf{R}(t; \widehat{\mathbf{X}}, \widehat{\mathbf{Y}}) - \mathbf{R}(0; \widehat{\mathbf{X}}, \widehat{\mathbf{Y}})| dt, \qquad (3)$$

where $T$ is the total duration of the degradation and the three reflectance color channels are red, $r$, green, $g$, and blue, $b$, for each sample, $(\widehat{\mathbf{X}}, \widehat{\mathbf{Y}})_n \in N$. High $I_c$ indicates high color change, corresponding to high degradation; $I_c$ close to zero indicates low color change and low degradation.

The performance of the degradation autocharacterization is demonstrated by comparing the output $I_c$ to the ground truth degradation, obtained from the pre- and post-band gap deviation[19,65] (Supplementary Fig. S-12a). Figure 7a illustrates the output of the autocharacterization where high computed $I_c$ values strongly corre-

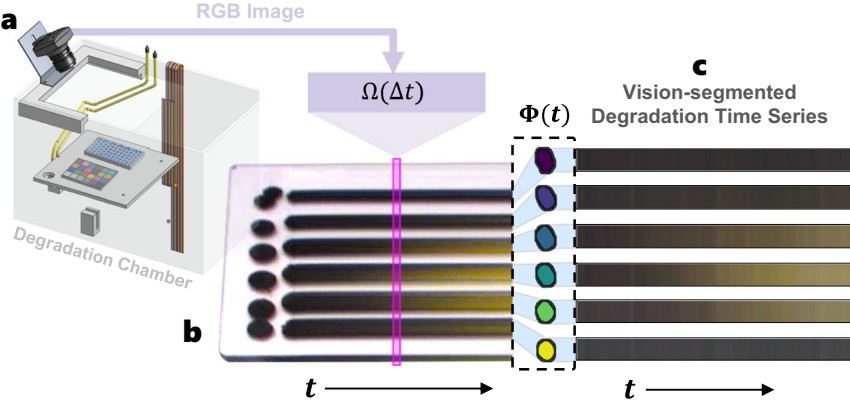

**Fig. 6 | Automated degradation testing and measurement of computer vision-segmented perovskite deposits. a** The samples are placed in the degradation chamber with specified environmental conditions for a total of 2 h. **b** RGB images of the samples are taken every 30 s for 2 h to resolve the time-dependent color change. The pink window indicates the pixels of a single image, **Ω**, taken during one-time step, Δ$t$, of the degradation procedure. **c** Computer vision is used to segment each deposited sample over time, **Φ**($t$), to compute the degradation intensity metric, $I_c$.

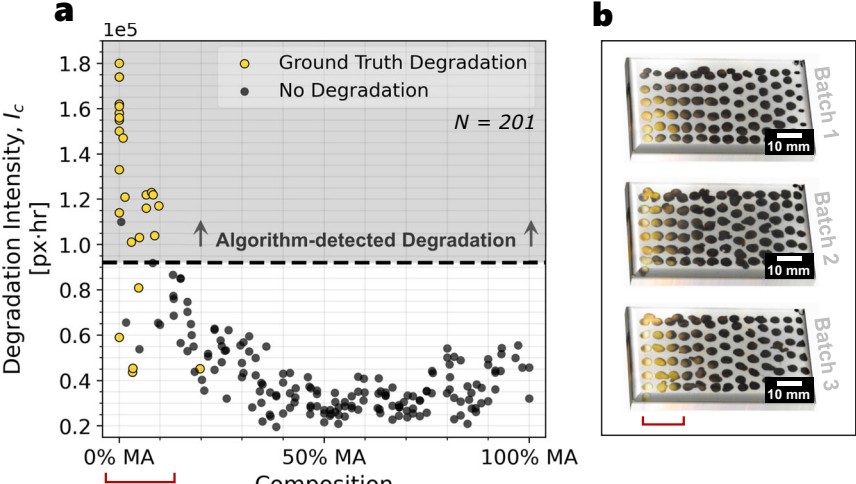

**Fig. 7 | Degradation detection autocharacterization algorithm performance compared against ground truth. a** The autocharacterization computed degradation intensity, $I_c$, relative to the ground truth degradation determined by a domain expert (yellow scatter points) on $N = 201$ unique perovskite samples across three independent trials. The black dashed line indicates the split between high and low $I_c$ values, corresponding to high and low degrees of degradation, respectively. **b** Images of the three batches of formamidinium (FA) and methylammonium (MA)

mixed-cation perovskite FA$_{1-x}$MA$_x$PbI$_3$ gradient samples after the 2-h controlled degradation. The leftmost samples are FA-rich and the rightmost samples are MA-rich. The yellowed FA-rich compounds, indicated by the red bracket, have undergone a phase transition from α-FAPbI$_3$ to δ-FAPbI$_3$. These compositions are considered as "ground truth" degradation samples if they exhibit a deviation of >0.02 eV in band gap from pre- to post-degradation, evaluated by a domain expert. Source data are provided as a Source Data file.

spond to the occurrence of the ground truth degradation in the samples (yellow scatter points). The determination of ground truth degradation is conducted by a human domain expert, further described in Supplementary Fig. S-12a. This classification performance of the autocharacterization algorithm achieves a precision-recall area under the curve (AUC) of 0.853 (Supplementary Fig. S-12c), and a maximal accuracy of 96.9%, relative to the ground truth (Supplementary Fig. S-12d). The yellowing pattern of the FA-rich samples is shown in Fig. 7b as a result of the phase change from favorable cubic phase $\alpha$-FAPbI$_3$ to the non-perovskite hexagonal phase $\delta$-FAPbI$_3$[63] (Supplementary Fig. S-13). Furthermore, running a full degradation detection computation using autocharacterization takes only 20 min/200 samples, given 48,000 total degradation images over the 2-h degradation experiment. This is a significant speedup from standard microscopy or XRD methods of determining degradation, which can take hours or days to identify the degradation of an equivalent number of samples.

Using the fast and accurate stability autocharacterization tool developed in this paper, we generate an ultra-high resolution stability trend for the FA$_{1-x}$MA$_x$PbI$_3$ series, shown in Fig. 7a where similar to band gap, this trend has not been reported at such a high resolution yet in literature to our knowledge. Prior literature reports stability compositional resolutions from $0 \leq x \leq 1$ for 11 compositions[66], 9 compositions[60], and 7 compositions[67] using conventional characterization methods. Moreover, Charles et al.[66] report the stability at $x \approx 0.1$ compositional resolution from $0 \leq x \leq 1$ using 6 timesteps, amounting to a total of 66 temporal data points. Comparatively, this study reports the stability at $x \approx 0.008$ compositional resolution from $0 \leq x \leq 1$ using 240 time-steps, amounting to 28,800 unique temporal data points (with 48,000 total temporal data points). Thus, with autocharacterization, we achieve over a 10× increase in the compositional resolution and a 40× increase in the temporal resolution for a total of a 436× increase in the number of unique data points reported for the FA$_{1-x}$MA$_x$PbI$_3$ stability series. Furthermore, with this high-resolution stability trend, we note the same regions of high-degradation appear in Fig. 7a as do in the literature for the $\alpha$-FAPbI$_3$ → $\delta$-FAPbI$_3$ degradation pathway at $0.0 \leq x \leq 0.15$, with the optimal low-degradation region occurring at $x \approx 0.40$[66,68]. Through the generation of ultra-high resolution trends, we may achieve a better understanding of complex semiconductor composition–property relationships to enable higher-performance design of materials in the future.

## Discussion

Accelerating the characterization of key material properties relevant to semiconductor engineering, such as band gap and stability, is a necessary step to enable the HT discovery and optimization of semiconductor materials. Conventional methods of characterization bottleneck the materials discovery pipeline when HT synthesis is utilized, inhibiting optimally efficient HT experimentation. For example, computing the band gap of 200 unique halide perovskite samples takes a domain expert over 8 h to complete.

In this work, we have demonstrated the fast and accurate characterization of band gap and detection of degradation within the FA$_{1-x}$MA$_x$PbI$_3$, $0 \leq x \leq 1$ perovskite system using parallelized and scalable computer vision segmentation. Through adaptive segmentation of inkjet-deposited materials, ultra-high resolution trends of band gap and degradation have been accurately and rapidly generated using the developed autocharacterization algorithms. The band gaps of 200 unique, HT synthesized perovskite samples were determined in 6 min at 98.5% accuracy within a 0.02 eV range using the band gap autocharacterization tool. Similarly, the degrees of degradation of 200 unique HT synthesized perovskite samples were determined within 20 min at 96.9% accuracy using the degradation autocharacterization tool.

Although we have developed the band gap autocharacterization algorithms for general use across various material compositions, in their present form, they are limited to the analysis of single-phase materials. Additionally, despite efforts to reduce systematic variance, the degradation experiments may include minor inconsistencies due to vibrations and lighting fluctuations that affect the computer vision signal for instability index calculations. Ensuring stringent control over degradation conditions remains crucial for reliable measurements. Future work aims to extend these algorithms to multi-phase materials, potentially featuring multiple band gaps, thereby enhancing the algorithms' versatility and general applicability in materials science research.

With the developed autocharacterization methods, we have demonstrated scalable and accurate characterization of band gap and stability properties of perovskite semiconductors at throughputs up to 85× faster than conventional domain expert evaluation. This advancement narrows the throughput gap between characterization and HT synthesis, taking a step toward debottlenecking the materials discovery pipeline. Therefore, the broader implementation of auto-characterization methods unlocks the potential to explore larger material search spaces faster, enabling the efficient design of higher performance functional materials.

## Methods
### Materials
3 in × 2 in × 1 mm glass slides (C&A Scientific) are cleaned using deionized water (DI, <1.0 μS cm$^{-1}$, VWR), Hellmanex III (VWR), and iso-propyl alcohol (IPA, ≥99.5%, VWR) to be used as substrates. Lead iodide powder (PbI$_2$, 99.999% trace metal basis, Sigma-Aldrich), formamidinium iodide powder (FAI, >99.9%, Greatcell Solar Materials), methylammonium iodide (MAI, >99.9%, Greatcell Solar Materials), dimethylformamide (DMF, ≥99.8%, Sigma-Aldrich), and dimethylsulfoxide (DMSO, ≥99.9%, Sigma-Aldrich) are used to prepare the perovskites.

### Computer vision segmentation of hyperspectral datacubes
A hyperspectral datacube of size $\mathbf{X} \times \mathbf{Y} \times \mathbf{R}(\lambda) \rightarrow 900\ px \times 800\ px \times 300$, where 300 is the number of wavelengths, $\lambda$, is captured as a raw image, $\mathbf{\Omega} = (\mathbf{X}, \mathbf{Y}, \mathbf{R}(\lambda))$, from the hyperspectral camera (Resonon, Pika L). This datacube is passed through several filters to find the edges and segment each material deposit sample $(\widehat{\mathbf{X}}, \widehat{\mathbf{Y}})_n \in N$ and then index the features appropriately, such that each pixel is mapped to its reflectance spectra, $\mathbf{R}(\lambda)$. Once segmented, each deposited material contains an area of ~1000 px worth of spatial spectral data. Inputting images or features of different sizes may require tuning the kernel sizes, $\kappa$, of the filters.

Algorithm 1 describes this segmentation process of $(\mathbf{X}, \mathbf{Y}, \mathbf{R}(\lambda)) \rightarrow (\widehat{\mathbf{X}}, \widehat{\mathbf{Y}}, \mathbf{R}(\lambda))$. First, the input image is cropped and converted to grayscale, then it is passed through several layers of thresholding and smoothing. By thresholding, eroding, blurring, and then thresholding again, we capture the edges of each feature while removing edge effects. The background is indexed as zeros, hence, all features split by zeros are assigned a unique index using island-finding graph network methods from the OpenCV Python library[52]: LabelFeatures(·) and Watershed(·)[53]. Once segmented, the features are smoothed, and any improperly segmented aberrations are pruned with the user-selected variables $\Theta_{min}$, $\Theta_{max}$, where features of size $s < \Theta_{min}$ or $s > \Theta_{max}$ are removed. Finally, a boolean mask is created for all pixels encoded with non-zero values to output $\mathbf{\Phi}$, where each uniquely indexed material deposit is directly mappable to the $\mathbf{R}(\lambda)$ measured for that deposit.

**Algorithm 1.** Scalable Computer Vision Segmentation

**Input** : $(\mathbf{X}, \mathbf{Y}, \mathbf{R}(\lambda))$: All image pixels and reflectance
$\Theta_{\min}$: Minimum segmented feature size
$\Theta_{\max}$: Maximum segmented feature size

**Output** : $(\widehat{\mathbf{X}}, \widehat{\mathbf{Y}}, \mathbf{R}(\lambda))$: Segmented pixels and reflectance

Let $\kappa$ be kernel size

```
1  img   ← (X, Y, R(λ))
2  img   ← Crop(img)
3  img   ← Grayscale(img)
4  img   ← 255 - img
5  mask  ← Binarize(img)
6  mask  ← MorphGradient(mask, κ = 12)
7  mask  ← Erode(mask, κ = 3)
8  mask  ← MedianBlur(mask, κ = 7)
9  mask  ← DistTransform(mask, κ = 3)
10 mask  ← LabelFeatures(mask)
11 (X̂, Ŷ)  ← Watershed(img, mask)
12 (X̂, Ŷ)  ← Dilate((X̂, Ŷ), κ = 5)
13 (X̂, Ŷ)  ← MedianBlur((X̂, Ŷ), κ = 7)
14 for Φ in Feature((X̂, Ŷ)) do
15    if Size(Φ) < Θ_min or Size(Φ) > Θ_max then
16       (X̂, Ŷ).Prune(Φ)
17    end
18 end
19 bool ← Boolean((X̂, Ŷ))
20 (X̂, Ŷ, R(λ)) ← (X, Y, R(λ)).Mask(bool)
   Return (X̂, Ŷ, R(λ))
```

## Composition mapping onto computer vision segmented data

The composition of each perovskite material is mapped onto each segmented deposit, $(\widehat{\mathbf{X}}, \widehat{\mathbf{Y}})_n \in N$, by corresponding each incremental pump speed and print head tool path coordinate to its associated time stamp. Once all pump speeds and path coordinates are time-stamped, they are mapped onto each other, and Eq. (1) is used to solve for $x$, the proportion of MA in the compound. This is done by first overlaying the print path onto the segmented image, $\mathbf{\Phi}$, containing each deposit, $(\widehat{\mathbf{X}}, \widehat{\mathbf{Y}})$. The raster path coordinates are acquired from the G-code controlling the printer tool path, where each coordinate has an associated time stamp. From there, each deposit gets assigned a time stamp range, $\Delta t = [t_a, t_b]$, based on the start and end time of path intersection. This maps each deposit to the time it was printed, $t(\widehat{\mathbf{X}}, \widehat{\mathbf{Y}})$. Then, the pump time steps are extracted directly from the microcontroller such that each set of pump speeds, $(\omega_{FA}, \omega_{MA})$, gets mapped to its corresponding time stamp, $t(\omega_{FA}, \omega_{MA})$. Finally, the time stamps assigned to each deposit coordinate get mapped to their respective pump time stamps $t(\widehat{\mathbf{X}}, \widehat{\mathbf{Y}}) \rightarrow t(\omega_{FA}, \omega_{MA})$. Since both pump speeds $\omega_{FA}$ and $\omega_{MA}$ are monotonic along the time series, they are uniquely deterministic of the proportion of FA and MA within the $FA_{1-x}MA_xPbI_3$ deposit by integrating pump speed over the range $\Delta t$ for each material deposit and solving for $x$ using Eq. (1).

## Band gap automation using hyperspectral data

The direct band gap for all $N = 201$ $FA_{1-x}MA_xPbI_3$ perovskite compositions is computed from the vision-segmented reflectance data, $\mathbf{\Phi} = (\widehat{\mathbf{X}}, \widehat{\mathbf{Y}}, \mathbf{R}(\lambda))$, since all the $FA_{1-x}MA_xPbI_3$ compositions are direct band gap materials at atmospheric pressure[69]. For every segmented sample, $(\widehat{\mathbf{X}}, \widehat{\mathbf{Y}})_n \in N$, the spatial median $\mathbf{R}(\lambda)$ spectra is used for computing the band gap. $\mathbf{R}(\lambda)$ spans across wavelengths, $\lambda$, where $\{\lambda \in \mathbb{Z} : 380\ nm \le \lambda \le 1020\ nm\}$ for hyperspectral imaging and $\lambda = \{r, g, b\}$ for the red, green, and blue color channels of RGB imaging. First, the

median $\mathbf{R}(\lambda)$ spectra are transformed into a Tauc curve using Eq. (2), with $\gamma = \frac{1}{2}$ for direct band gap. Then, transformed Tauc curves are recursively segmented in half until each segment achieves a fit of $R^2 \ge 0.990$, indicating that each segment is near-linear:

$$R^2 = 1 - \frac{\sum_i^N (y_i - \widehat{\mathbf{Y}}_i)^2}{\sum_i (y_i - \bar{y})^2}, \quad (4)$$

where $\widehat{\mathbf{Y}}$ is the predicted value and $\bar{y}$ is the average value of the set. Once the recursion is complete, each pair of adjacent line segments is iteratively fit to a linear regression line, generating a set of candidate fit lines to use for computing band gap. To determine the best candidate fitted line, root-mean-square error (RMSE) is used rather than using the inclination angles of Tauc curves[34] to improve generalizability across different materials, e.g., $FAPbI_3$ and $MAPbI_3$:

$$RMSE = \sqrt{\frac{\sum_i^N (y_i - \widehat{\mathbf{Y}}_i)^2}{N}}. \quad (5)$$

We implement an iterative RMSE minimization routine that automatically identifies the Tauc curve peaks to fit between. Then, the RMSE is computed between each regression line and the Tauc curve within the lower bound of the regression $x$-intercept and the upper bound of the Tauc peak location minus one-half of the peak width. Enforcing the RMSE computation to occur within these bounds was shown to increase fitting accuracy with the Tauc slope. The band gap, $E_g$, is then extracted from $x$-intercept point of the regression line that achieves the minimum RMSE.

## Detecting perovskite degradation from RGB time series data

In order to use color as a reproducible and repeatable quantitative proxy for degradation, color calibration needs to be applied because the illumination conditions in the aging test chamber may create distortions to the true sample color. At the beginning of the degradation study, an image of a reference color chart (X-Rite Colour Checker Passport; 28 reference color patches), $I_R$, is taken under the same illumination conditions as the perovskite semiconductor samples. Images at each time step, $\mathbf{\Omega}(\Delta t)$, are transformed into CIELAB colorspace and subsequently into a stable reference color space, CIE 1931 color space with a 2-degree standard observer and standard illuminant D50, by applying a 3D-thin plate spline distortion matrix $\mathbf{D}$[19,70] defined by $I_R$ and known colors of the reference color chart:

$$\mathbf{D} = \begin{bmatrix} \mathbf{V} \\ \mathbf{O}(4,3) \end{bmatrix} \begin{bmatrix} \mathbf{K} & \mathbf{P} \\ \mathbf{P}^T & \mathbf{O}(4,4) \end{bmatrix}^{-1} \quad (6)$$

Here, $\mathbf{O}(n, m)$ is an $n \times m$ zero matrix, $\mathbf{V}$ is a matrix of the color checker reference colors in the stable reference color space, $\mathbf{P}$ is a matrix of the color checker RGB colors obtained from $I_R$, and $\mathbf{K}$ is a distortion matrix between the color checker colors in the reference space and in $I_R$. Using the color-calibrated images and droplet pixel locations given by $\mathbf{\Phi}$, a final array, $\mathbf{R}(t; \widehat{\mathbf{X}}, \widehat{\mathbf{Y}})$ of the average color at time $t$ for perovskite semiconductor of composition $FA_{1-x}MA_xPbI_3$ is created. The color of each droplet is measured to determine the stability metric $I_c$[19], calculated using Eq. (3).

## Experimental section

**Substrate preparation.** Glass slide substrates are prepared for printing the perovskite samples using a three-step cleaning process: (1) ultrasonication for 5 min in DI water with 2% vol. Hellmanex III solution, (2) ultrasonication for 5 min in DI water only, and (3) ultrasonication for 5 min in IPA. Once cleaned, the substrates are transferred to an inert nitrogen environment glovebox with moisture levels <10 ppm.

**Perovskite preparation.** $FAPbI_3$ and $MAPbI_3$ are prepared as 0.6 M liquid-based precursors for HT printing. For printing, 2 mL of each precursor is prepared in an inert nitrogen environment glovebox with moisture levels <10 ppm. First, 3.2 mL DMF is mixed with 0.8 mL of DMSO to make 4 mL of 4:1 DMF:DMSO solution. Then, 1.106 g of $PbI_2$ powder is dissolved into the 4 mL of 4:1 DMF:DMSO to make a $PbI_2$ stock. Next, the 4 mL $PbI_2$ stock is split in half, pipetting 2 mL of stock/vial. Lastly, 0.206 g of FAI powder is dissolved into one of the 2 mL $PbI_2$ stock vials and 0.191 g of MAI powder is dissolved into the other making 0.6 M $FAPbI_3$ and 0.6 M $MAPbI_3$, respectively.

**High-throughput perovskite synthesis.** The liquid-based $FAPbI_3$ and $MAPbI_3$ precursor solutions are deposited using an inkjetting, HT combinatorial printer, synthesizing $N = 201$ unique $FA_{1-x}MA_xPbI_3$ composition samples[18]. To begin printing, first, all printer plumbing lines are flushed twice with the 4:1 ratio DMF:DMSO solution. Then, the $FAPbI_3$ and $MAPbI_3$ precursors are extracted into syringes using a microcontroller to communicate with the pumps. These syringes contain prismatic motion plungers that use positive displacement to fill and eject the solution. Next, all plumbing lines are primed with the precursor solution. After priming, the precursors are purged at equal rates from both syringes for 50 s to remove air bubbles. Finally, motor encoders pump the precursors out of the syringes at pre-programmed rates, illustrated in Supplementary Fig. S-3, and enter a mixing chamber prior to deposition. A pinch valve breaks up the fluid flow within a 1/32 in inner diameter × 3/32 in outer diameter silicone tube, actuating at 11 Hz frequency and 5% duty cycle to deposit each sample as a discrete droplet onto the cleaned glass slide. The print head translates at a speed of 38 mm s$^{-1}$ over the 3 in × 2 in glass slide in a serpentine pattern, depositing ~70–80 unique composition samples per batch in 16.5 s. Approximately 0.15 mL of total precursor volume is consumed per print, which includes the volume required for purging and priming the plumbing lines. After the droplets have been deposited onto the substrate, the substrate is transferred to a hotplate to anneal for 15 min at 150 °C.

### Reporting summary
Further information on research design is available in the Nature Portfolio Reporting Summary linked to this article.

## Data availability
The hyperspectral data and band gap generated in this study have been deposited in the OSF database under accession code https://osf.io/qe9ax. The degradation data generated in this study have been deposited in the OSF database under accession code https://osf.io/y2xds. Source data are provided with this paper.

## Code availability
All codes used to develop the autocharacterization and computer vision algorithms are available publicly with complete working examples. The band gap autocharacterization code (https://github.com/PV-Lab/Autocharacterization-Bandgap) and the degradation detection autocharacterization code (https://github.com/PV-Lab/Autocharacterization-Stability) are available from GitHub.

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

## Acknowledgements

We thank First Solar for support and fruitful discussions. The automation portion of this work was undertaken thanks in part to funding provided to the University of Toronto's Acceleration Consortium. This material is partially based upon work supported by the U.S. Department of Energy's Office of Energy Efficiency and Renewable Energy (EERE) under the Advanced Manufacturing Office (AMO) Award Number DE-EE0009096. B.D. acknowledges the support of the Simons Foundation. A.T. was supported by the Research Council of Finland Flagship programme: Finnish Center for Artificial Intelligence (FCAI). H.K. received financial support from the Scientific and Technological Research Council of Turkey (TÜBİTAK) BİDEB-2219 (Project No: 1059B192100703) during this study. This work made use of the MRSEC Shared Experimental Facilities at MIT, supported by the National Science Foundation under award number DMR-1419807.

## Author contributions

A.E.S. and E.A. conceptualized the work. A.E.S., E.A., A.T., B.D., and T.B. designed the methodology. A.E.S., E.A., and A.T. wrote the software. F.S. and H.K. prepared the experimental materials. A.E.S., E.A., F.S., and B.D. conducted experiments. A.E.S. and E.A. performed the analysis. H.K. performed expert human benchmarking. A.E.S. and E.A. wrote the manuscript. All authors reviewed and edited the manuscript. A.T., B.D., and T.B. provided guidance.

## Competing interests

The authors declare no competing interests.
