## [Peer Review File · Nature Communications]

Using Scalable Computer Vision to Automate High-throughput Semiconductor CharacterizationREVIEWER COMMENTS

Reviewer #1 (Remarks to the Author):

The work deals with the high speed characterization of thin film halogenated materials for photovoltaic applications. In particular it explains the method developed and used for high throughput characterization from hyperspectral data and computing vision. It is significant for the field since perovskite composition can be quite complex and high throughput can be easily used to screen this composition; thus high speed characterization is needed.

Two main remarks from my review are :

(1) how is the reproducibility of the methods for the same samples? it is not shared in the paper.
(2) how can we deal with inhomogeneous deposition of the film and thus inhomogeneous ageing of the thin film? in the paper the analyses on degradation are quite simple in comparison of what happens in reality (dots of degradation, coffee ring, thickness variation and thus various degradations,...)

Reviewer #2 (Remarks to the Author):

Overall recommendation

This paper makes a valuable advance in accelerated discovery of a specific class of materials. It may be suitable for Nature Communications, with suitable expansion of the argument to show its broader applicability. At the moment, this is not immediately clear and reads as being most immediately valuable to a more specialized audience. Parts of the methodology are, in addition to the valuable references acknowledged by the authors, preceded elsewhere. For example, DOI: 10.1039/D2DD00089J reports the use of time-resolved computer vision to characterize materials degradation in a high throughput environment. To support the authors' revision of the manuscript, whether or not it is published in this journal, I have provided additional notes in the attached PDFs.

Summary

The authors have described a new computer-vision enabled, automated method of characterizing material properties (specifically Perovskite-based materials). A composition mapping tool is demonstrated for materials that are synthesized from two parts. They have demonstrated a system that overcomes the bottlenecks associated with the band gap analysis of Perovskite-based semiconductor materials. Currently, this is a fairly labor intensive process and requires a lot of human intervention.

Abstract

This was constructed in a clear and concise manner and communicated the overall point of the paper well.

Introduction

The introduction has been written in a rather verbose fashion. It made it difficult to follow in parts. A partial rewrite is needed here if a broad audience is to be served.

The phrasing could be more concise e.g.

"The importance of developing rapid and accurate methods of characterization for HT materials discovery and optimization derives from the intractable nature of exhaustively testing every material within a functional material's search space using these conventional tools [14]."

Also, in the above quote, I believe 'intractable' is used incorrectly.

The authors state: "For one to exhaustively search this subspace at 1% compositional steps would

require synthesizing and characterizing 7×10^{12} unique material samples (Supplementary Figure S-14).”

And then state:

“Thus, only small regions of this search space can be explored experimentally with current methods, given the large discrepancy between search space size and characterization throughput”

The above isn't strictly true, given the large number of unique materials one could either look at small regions of a search space or sparsely look at many regions.

The selection of papers displaying the current state of the art is commendable and are good example of holes in the art in producing automated processes for analyzing band gaps. However, the aims of the authors' current work seem somewhat lost in this. They state that only small regions of the search space can be explored, then give examples of bottlenecks in analysis.

Overall, the link between reducing bottlenecks which then leads to the ability to synthesize and analyze more materials within a given design-space, could be better articulated.

With that being said, the stated content of the paper at the end of the introduction is quite clear.

Results

The results section was relatively clear and mathematically robust. One point that would improve the statistical analysis would be to ensure the reader that the regression analysis is being performed on data for which regression is applicable, i.e. show that the data are not fat-tailed.

Conclusions

There should be some inclusion of the limitations of the new composition mapping tools. There is some mention of repeatability in the SI but more of this should come through in the conclusions.

SI

Overall, the SI seems well presented and includes the level of detail one should expect.

Further notes to help the authors improve the manuscript can be found on the uploaded manuscript and SI documents.

Response to Reviewer's Comments:

The authors would like to thank the editor and peer reviewers for their insightful comments and suggestions. The changes suggested by the reviewers have been implemented into the text body as well as the figures presented in the paper. We believe that these changes have significantly improved both the quality and clarity of the manuscript. Below we address each point by the reviewers (the reviewer's comments are in blue, specifically within brackets). For comments made directly in the manuscript PDF, we provide the complete sentence for context and then highlight the segments marked by the reviewer. Comments made by the reviewer regarding the highlighted segments are placed in brackets.

Reviewer #1 Comments to Authors:

1. [how is the reproducibility of the methods for the same samples? it is not share in the paper.]

Response:

We would like to thank the reviewer for their comment regarding reproducibility across samples. We hold reproducibility of our results to a very high standard and try to extensively indicate how our methods perform across unique batches of materials. Firstly, we demonstrate the reproducible performance of both the band gap and degradation computation on three unique batches of materials. The scatter plots in the manuscript, specifically Figure 5 and Figure 7a, show the individually plotted points across all three of these batches for band gap and degradation, respectively. Furthermore, it is noted in both captions of the aforementioned figures that these scatter points are “unique perovskite samples across 3 independent trials.” Each trial consists of approximately 70 uniquely synthesized samples from the $\text{FA}_{1-x}\text{MA}_x\text{PbI}_3$, which is stated in the Methods section, “High-throughput Perovskite Synthesis” on page 17 of the manuscript.

Figure S-7a

Figure S-12b

In addition to the above statements on reproducibility within the main manuscript, we further illustrate which batches each sample comes from in Supplementary Information (SI). Specifically, Figure S-7a and Figure S-12b in the SI (reproduced above for convenience) discretize each sample based on the batch they were produced in to further highlight reproducibility between batches. For example, we can see that batches 1 and 3 have strongly corresponding band gaps while batch 2 deviates along the FA-rich

compounds, likely due to degradation of samples rather than algorithmic performance.

Hence, experimental reproducibility of the samples limits perfect reproducibility of autocharacterization algorithm performance across batches. We further dive into the limitations of experimental reproducibility of samples produced using high-throughput ink-based synthesis in the section of the SI entitled “Experimental Reproducibility” (page 2 of the SI). In this section, we highlight that morphological variations sometimes arise as a result of the high-throughput synthesis process, such as imperfect grain boundaries, accompanied with optical microscopy data. Moreover, we state “These morphological variations in the high-throughput manufacturing process must be further studied to ascertain better control over experimental reproducibility.”

2. [how can we deal with inhomogeneous deposition of the film and thus inhomogeneous ageing of the thin film? in the paper the analyses on degradation are quite simple in comparison of what happens in reality (dots of degradation, coffee ring, thickness variation and thus various degradations,...)]

Response:

This is a wonderful question posed by the reviewer. The reviewer is correct in stating the difficult reality of inhomogeneous degradation of perovskites, which arises due to a multitude of factors. To dissect the performance of the autocharacterization algorithms on inhomogeneous degradation, we have added an additional analysis in the “Stability” section of the SI. This analysis can be found on pages 8-9 of the SI and Figure S-10 of the SI (figure reproduced below for convenience).

Figure S-10

By default, the stability autocharacterization algorithm computes the instability index, I_c , as a spatial average across all pixels. Instead of computing the spatial average, Figure S-10 shows the I_c computation per pixel for every pixel in the three called out samples. Each line indicates the I_c values for a given pixel and the color corresponds to the magnitude of degradation, yellow indicating higher degradation. The first sample is fully degraded, the second sample has spotted degradation, and the third sample is not degraded. Here, I_c is computed for each time step in the degradation experiment. We can see that the spotted degradation results in the I_c values being very spread out by the final time step since some spots in the sample are highly degraded while others are not degraded.

Although the per-pixel I_c values may differ across a sample with spotted degradation, we can see that by

taking the spatial average of these values, as indicated by the red dashed line, degree of degradation is still well-determined by the algorithm. Comparing the red dashed line for the three samples, we can see that the degraded and spot-degraded samples both produce higher spatially averaged I_c values than the non-degraded sample.

Reviewer #2 Comments to Authors:

1. [This paper makes a valuable advance in accelerated discovery of a specific class of materials. It may be suitable for Nature Communications, with suitable expansion of the argument to show its broader applicability. At the moment, this is not immediately clear and reads as being most immediately valuable to a more specialized audience.]

Response:

We thank the reviewer for their comment and strongly agree with the reviewer on their assessment of necessity for broader applicability. In the prior iteration of the manuscript, the introduction was written verbosely with a tone directed more for a specialized perovskite semiconductor audience. In the newest iteration of the introduction and discussion, we have implemented all of the comments posed by the reviewer below to improve the manuscript's broader applicability to include the entirety of the automated and high-throughput materials discovery community. With this revision of the introduction and discussion, we believe broader applicability of the methods is highlighted more clearly.

2. [Parts of the methodology are, in addition to the valuable references acknowledged by the authors, preceded elsewhere. For example, DOI: 10.1039/D2DD00089J reports the use of time-resolved computer vision to characterize materials degradation in a high throughput environment.]

Response:

We thank the reviewer for indicating some of the prior work which does support the foundational methodology of the current paper. The paper pointed out by the reviewer, published in RSC, is one of our papers, co-written by authors A.T, A.E.S, and T.B. who are on the current paper in discussion. In the RSC paper, we illustrate how to build low-cost high-throughput degradation chambers and discuss several different methods of degradation detection across thin films, bulk crystals, and full photovoltaic devices. In the RSC publication, bulk crystals are placed onto white circular paper cutouts by the researcher such that simple thresholding can be used to find bulk crystal placement. Cutting out white paper circles and placing them beneath the samples is feasible $n = 9$ samples, which is the number of samples that paper analyzes. However, this is infeasible for $n = 201$ samples, which is the number of samples the current paper in review analyzes. Moreover, these rudimentary vision methods in the RSC publication require calibrating a "blank" well containing the bulk crystals. Therefore, in order to expand the generalizability and accelerate the throughput, the current paper in discussion has been designed to include several layers of filters and more sophisticated edge detection to enable the adaptive vision segmentation of complex sample patterns containing over 70 uniquely deposited materials synthesized using high-throughput synthesis tools. Furthermore, we have designed the new computer vision tools for parallel computing of material properties while also being able to handle unknown positions of samples with varying morphologies.

3. [The introduction has been written in a rather verbose fashion. It made it difficult to follow in parts. A partial rewrite is needed here if a broad audience is to be served.]

The phrasing could be more concise e.g.

“The importance of developing rapid and accurate methods of characterization for HT materials discovery and optimization derives from the intractable nature of exhaustively testing every material within a functional material’s search space using these conventional tools [14].”

Also, in the above quote, I believe ‘intractable’ is used incorrectly.]

Response:

As mentioned in Review #2 Response #1, we have conducted a major re-write of the introduction section of the paper, following the reviewer feedback to expand applicability and improve conciseness. These changes can be seen in the Changes PDF file, highlighted in orange. The sentence called to attention by the reviewer has since been removed in the rewrite.

4. [The authors state: “For one to exhaustively search this subspace at 1% compositional steps would require synthesizing and characterizing 7×10^{12} unique material samples (Supplementary Figure S-14).”

And then state:

“Thus, only small regions of this search space can be explored experimentally with current methods, given the large discrepancy between search space size and characterization throughput”

The above isn’t strictly true, given the large number of unique materials one could either look at small regions of a search space or sparsely look at many regions.]

Response:

We agree with the reviewer that the prior claims are not always true. In the current re-write, both sentences have been removed in favor of more concise statement, “Optoelectronic materials, such as semiconductors, highlight this challenge because of their complex and extensive material search space.”

5. [The selection of papers displaying the current state of the art is commendable and are good example of holes in the art in producing automated processes for analyzing band gaps. However, the aims of the authors’ current work seem somewhat lost in this. They state that only small regions of the search space can be explored, then give examples of bottlenecks in analysis.]

Overall, the link between reducing bottlenecks which then leads to the ability to synthesize and analyze more materials within a given design-space, could be better articulated.]

Response:

We appreciate the positive comment regarding our selection of prior art papers. To better strengthen the link between our aims and the provided examples, we have reduced the focus on the scale of the design space in favor of keeping the focus on the bottlenecks. Hence, we highlight the high-throughput synthesis tools and how they further drive a wider wedge between characterization and synthesis throughputs. This message more closely aligns with the examples provided within the selection of papers, which we believe improves the cohesion of the sections. These changes are also highlighted in orange in the Introduction.

6. [The results section was relatively clear and mathematically robust. One point that would improve the statistical analysis would be to ensure the reader that the regression analysis is being performed on data for which regression is applicable, i.e. show that the data are not fat-tailed.]

Response:

To address the reviewer’s comment on the relevance of regression analysis on the data, we provide the requested histogram of data. The addition of this analysis can be found on page 6 of the SI. In Figure S-6 of the SI (reproduced below for convenience), we show a histogram highlighting the distribution of reflectance values at the band gap used for regression fitting in the paper. This histogram shows data for pixel spectral data across all materials synthesized in the study. The distribution is noted to be normal without any fat tails.

Figure S-6

7. [There should be some inclusion of the limitations of the new composition mapping tools. There is some mention of repeatability in the SI but more of this should come through in the conclusions.]

Response:

We have added an explicit limitations paragraph to the Discussion section, now found on page 13 of the manuscript.

8. In reference to the highlighted segment from the Abstract:

However, with the boom in development of high-throughput synthesis tools that champion production rates up to 10^4 samples per hour with flexible form factors, most sample characterization methods are either slow (conventional rates of 10^1 samples per hour, approximately 1000x slower) or rigid (e.g., designed for standard-size microplates), resulting in a bottleneck that impedes the materials-design process.

[Is there a better way to express this?]

Response:

We have changed the wording from “flexible form factors” to a more clear and explicit wording of “variable morphologies.” Each droplet deposited using the ink-based synthesis methods is slightly different from the rest, mostly in its geometric parameters and placement, in turn, affecting morphology. Thus, we believe “variable morphologies” more closely captures this sentiment.

9. In reference to the highlighted segment from the Introduction:

To discover commercially relevant semiconductor materials, e.g., for solar applications [1-3], vast compositional search spaces must be rapidly synthesized and characterized, e.g., for band gap [4-6] and stability [7, 8].

[verbose]

Response:

As noted in the response to Reviewer #2 Responses #1 and #5, we have deemphasized the focus on design space in favor of a stronger and clearer focus on the synthesis-characterization bottlenecks. As a result, this sentence has been removed from the introduction.

10. In reference to the highlighted segment from the Introduction:

Although these HT manufacturing methods have shown great progress in driving the rapid screening of large material search spaces in an automated fashion, much of the materials characterization process is still hindered due to its manual nature [14, 19] or rigid microplate-based form factors [9, 10, 20-22].

[Could this be rephrased?]

Response:

We have rephrased this sentence to the following: “Consequently, these droplet morphologies are not readily compatible with existing characterization tools, which are designed for uniform thin film analysis.” We focus on characterization tools designed for standard thin films instead, as this is more broadly applicable.

11. In reference to the highlighted segment from the Introduction:

The metal halide perovskite material search space is both highly dimensional and vast, hence, as a result, it is intractable to map using conventional synthesis or characterization methods.

[The following section on the current state of the art could be much clearer. The authors have done well to collate examples of gaps in producing automated processes for analysing band gaps. However, the aims seem somewhat lost. They state that only small regions of the search space can be explored, then give examples of bottlenecks in analysis.]

Response:

We thank the reviewer on their acknowledgement of our collection of examples. As noted in Reviewer #2 Response #5, we remove focus from the message of the design space in favor of highlighting the bottlenecks and their relevant background. We believe this improves the cohesion between the aims and

the provided examples. As a result, we have conducted a major re-write of this paragraph in the introduction, as indicated by the orange highlights of the Changes PDF.

12. In reference to the highlighted segment from the Introduction:

Thus, only small regions of this search space can be explored experimentally with current methods, given the large discrepancy between search space size and characterization throughput.

[This is not true. You could search large areas of the search space, albeit very sparsely.]

Response:

This statement has been removed from the manuscript.

13. In reference to the highlighted segment from the caption of Figure 3 in the Results:

b, XRD peak traces at the (012) crystallographic plane measured at uniformly spaced compositions in the batch print.

[Not sure what this means...]

Response:

To improve the clarity of this notation, in the caption of Figure 3 in the results, we have changed “XRD peak traces at the (012) crystallographic plane measured at . . .” to “XRD peak traces at the crystallographic plane with the Miller indices (012) measured at . . .” This more clearly defines the notation of (012) to be the Miller indices of the mentioned crystallographic plane.

14. In reference to the highlighted section from the Methods:

Perovskite Elemental Composition Mapping onto Computer Vision Segmented Data

[This section could be written in a way that is clearer. Describing the pump speeds as both variable and monotonic in the same sentence gives the impression of a contradiction.]

Response:

We agree with the reviewer that this section required a re-write to improve clarity. To address this comment, we have conducted a major re-write of this section within the methods section. Furthermore, we no longer describe the pump speeds as “variable,” instead only as “monotonic.”

REVIEWERS' COMMENTS

Reviewer #1 (Remarks to the Author):

The answers are convincing.
The paper can be published.

Reviewer #1 (Remarks on code availability):

The answers are convincing.
The paper can be published.

Reviewer #2 (Remarks to the Author):

Siemenn and co-workers have done an exceptional job of answering and actioning all comments raise by all reviewers.

As noted further in the appropriate section below, I also highlight the excellent code quality submitted by the authors and made available on the cited repository.

I now fully support this article for publication in its current form.

Reviewer #2 (Remarks on code availability):

The code repository, both in terms of the README page and structured commenting of the code itself, is robustly documented. The scripts demonstrate adequate cohesion to enable maintainability and adoption of the code into other workflows.

This reviewer finds no issues and commends the authors for maximising the re-use value of their software.